# The Relationship between Work-Related Stressors and Construction Workers' Self-Reported Injuries: A Meta-Analytic Review

**Bandar Moshabab Alqahtani** [1,*]**, Wael Alruqi** [2]**, Siddharth Bhandari** [3]**, Osama Abudayyeh** [4] **and Hexu Liu** [4]

1    Department of Civil and Construction Engineering, College of Engineering,
     Imam Abdulrahman Bin Faisal University, Dammam 31451, Saudi Arabia
2    Civil Engineering Department, College of Engineering, Jouf University, Sakaka 72388, Saudi Arabia
3    Research Faculty and Associate Director of Research of Construction Safety Research Alliance,
     Department of Civil, Environmental, and Architectural Engineering, University of Colorado Boulder,
     1111 Engineering Drive, UCB 428, Boulder, CO 80309-0428, USA
4    Department of Civil and Construction Engineering, Western Michigan University,
     Kalamazoo, MI 49008-5316, USA
*    Correspondence: bmalqahtani@iau.edu.sa

**Abstract:** Mental health concerns are surging worldwide and workers in the construction industry have been found to be particularly vulnerable to these challenges. Stress, depression, addictions, suicides, and other key indicators of poor mental health have been found to be highly prevalent among construction workers. Critically, researchers have also found a link between how stress in the workplace impacts the overall safety performance of an individual. However, the burgeoning nature of the research has stifled the determination of feasible and actionable interventions on jobsites. This paper aims to analyze the relationship between work-related stressors found on construction jobsites and self-reported injury rates of workers. To accomplish this goal, a meta-analysis methodology was used, wherein a comprehensive literature search was conducted to identify key work-related stressors and questionnaires used in the construction industry's safety domain to assess stress. Using a formal meta-analysis approach that leverages the findings from past studies, a more holistic determination of the relationship between work-related stressors and injury rates among workers was performed. Ninety-eight studies were reviewed, and seven were selected that fulfilled pre-determined validated inclusion criteria for eligibility in the meta-analysis. The results revealed 10 salient work-related stressors among construction workers. Of these ten, seven work-related stressors were identified as significant predictors of injury rates among workers: job control, job demand, skill demand, job certainty, social support, harassment and discrimination, and interpersonal conflicts at work. This study represents a significant first step toward formally identifying work-related stressors to improve working conditions, reduce or eliminate injuries on construction sites, and support future research.

**Keywords:** work-related stressors; self-reported injuries; stress in construction; job stressors; physical stressors; organizational stressors; work stress; job control; job certainty; meta-analysis; stress

## 1. Introduction

The growing list of safety- and health-related concerns of employees within the construction industry is well-established and unfortunate. Until recently, academics and professionals in the field have focused primarily on the physical well-being and safety of workers on construction sites. With recent research showing that mental health problems have not only caused productivity losses but also raised safety concerns [1], more attention is being paid to the health of employees on jobsites. With the construction industry reporting higher levels of mental health issues than industries in general [2], it is unsurprising that the industry has been struggling to recruit and retain skilled workers across the globe [3,4].

These health-related concerns are not unique to a particular sector or demographic within the industry but are part of a global trend [5]. Data from the United Kingdom and the United States of America shows that the risk of suicide among low-skilled construction workers is higher than the national averages [6,7], and the construction industry in Australia shows higher substance-abuse rates than other industries [8]. These trends have prompted a recent but significant interest from academics and practitioners in developing knowledge and resources in understanding [9,10], measuring [11–13], and responding to [14] mental health challenges on construction sites. It is critical for the sustainability of the industry, on a global scale, to address mental health challenges with urgency.

Researchers have found that stress levels of individual workers can influence productivity, safety performance, and perceptions toward organizations within the construction domain [15]. Several antecedents to stress in the workplace, including but not limited to work overload [16,17], lack of autonomy [16], inadequate compensation [18], and values [19] have been shown to directly or indirectly facilitate risky situations and behaviors. The predictive nature of stress on the risk-taking tendencies of workers is also well-documented [20,21]. However, most of these studies used different stress surveys with varying definitions that did not provide a singular agreed-upon epistemological position for the construction of the survey. This makes comparing and evaluating the validity of the results a laborious exercise. The answer to the question what work characteristics are causing increased mental and physical stress on workers? still needs more edification.

This paper aims to formally investigate the relationship between work-related stressors and construction workers' self-reported injuries, thereby identifying the essential work-related stressors that influence safety performance on jobsites. A formal statistical meta-analysis was conducted on the dimensions of stress to leverage the statistical knowledge from multiple published studies for the purpose of facilitating comparisons and aggregations. Through such an explorative statistical exercise, not only were the authors able to determine the salient dimensions of stress for construction sites in context, but also to show the capacity of those stresses in predicting injury rates. Within the construction safety literature, a meta-analysis focused on workplace stresses has not previously been conducted. Although researchers from other domains have conducted such explorations [22,23], those findings cannot be generalized to the unique working environments and workforce demographics found on construction jobsites. It has also been demonstrated by past statistical meta-analyses of stress performed in different disciplines that a consistent list of stressors may not be possible for all occupational contexts and conditions [22,23]. Therefore, this paper's exploration will allow safety researchers and industry practitioners to develop new targeted interventions on construction jobsites to improve the physical and mental well-being of their employees.

Specifically, we sought to achieve the aforementioned research goal by (1) examining review studies and questionnaires that measure stress; (2) identifying the key dimensions of stress; (3) establishing a consistent definition for each dimension; and (4) performing a meta-analysis to establish the relationship between individual dimensions of stress and self-reported injury rates. We used peer-reviewed publications within the construction sector from 1989 to 2020 to establish the relationship between the dimensions of stress and safety performance. This approach to meta-analysis has been previously validated for the construction sector by Alruqi et al., 2018 [24].

## 2. Literature Review

A literature search was conducted to assemble and code all studies related to work-related stressors and self-reported injuries in the construction industry. First, a search using a large-scale range of single or combined keywords in several known sites, such as Google Scholar, ResearchGate, Scopus, Engineering Village, PsychInfo, and the American Society of Civil Engineers, was carried out. Second, to develop the keywords, our objective was to construct a list of words or phrases as specific as possible to the phenomena under investigation and to encompass the range of words used to represent the phenomena [25]. The

keywords that we used were "construction," "stress in construction," "job stressors", "job stress", "Injuries", "interpersonal conflict", "safety performance", "individual resilience", "task stressors", "organizational stressors", "personal stressors", "interpersonal stressors", "physical stressors", "emotional stress", "physical stress", "work stress", "supervisor support", "co-worker support", "job certainty", and "job control". The search resulted in identifying 98 previous studies.

## 3. What Is Stress?

There is no universal definition of "stress" in the current literature. Stress is deeply ingrained into the everyday vernacular, which makes it difficult to establish a singular ontological position around it. It is a challenging construct to pin down from a definitional point of view because we have different types of stresses being experienced by different entities (e.g., individual, medical, organizational, and environmental). Selye, 1973 [26] proposed that stress could be defined as "the nonspecific response of the body to any demand made upon it." Stress is induced and triggered by external demands (e.g., work, spouse, parental, political, technological, and societal) or internal causes (e.g., shame, anxiety, and depression), wherein the stressed individual is working under a growing sense of uncertainty about desired outcomes [27]. In this research, we adopted a generalized definition that grounds our epistemological position: stress is a negative force applied to an individual from the surrounding environment [28] that leads the individual to interact with that particular situation [17].

Stress can be both personal and work-related. Personal stress is experienced by an individual by being in situations, conditions, or events that hinder desired outcomes in life and negatively impact the health or well-being of the individual and/or his/her loved ones [29]. On the other hand, work-related stresses result from situations, conditions, or events that are directly attributable to an individual's place of work and that cause physiological, psychological, and behavioral responses [30]. Within the construction safety context, work-related stress can be characterized as the experience of psychological and physiological negative reactions resulting from perceived or existing unpleasant, unexpected, or immediate workplace risks [31]. This study focused on work-related stress specifically, as the interactions between personal stress and safety performance have not been adequately considered in the literature, across different contexts, to support a formal meta-analysis.

## 4. Stressors in the Construction Industry

The definition of stressors describes perturbations that could disrupt an organism's optimal functioning [32,33]. In simple words, stressor is the label for the unique antecedents to stress (i.e., the causes of stress). Researchers have found a number of work-related stressors, such as lack of job security, lack of job satisfaction, working in a dangerous environment, and poor workgroup relationships that adversely impact individuals' job performance, decision-making skills, and behavior [34,35]. Furthermore, research has shown that stressors can negatively influence learning outcomes and motivations [36], including those of adult learners, especially in non-traditional learning environments [37,38]. Thus, the assessment and management of salient stressors are of paramount importance in enhancing not only safety performance but also well-being within the workforce in the construction industry.

Within the construction engineering and management (CEM) domain, a number of approaches have been used to measure the stress levels among workers, including but not limited to the following: (1) interview questions (e.g., [39–41]), (2) questionnaire surveys (e.g., [11,15,42–44]), and (3) focus groups (e.g., [45,46]). To enhance external validity, most studies have relied on self-reporting questionnaires with a Likert scale, which allow participants to self-report the extent to which they have experienced different stressors. The ratings across different stressors were then used to produce an aggregated stress score. This approach was used in approximately 31% of the 98 studies we reviewed (e.g., [47–51]).

Therefore, most CEM literature studies represent quasi-field experiments with limited internal validity, due to the practical challenges of conducting a controlled experiment to measure stress levels among workers. Finally, of the 98 studies reviewed, 97 focused on homogenous samples, and only one examined and compared construction professionals' stress levels between two countries [52]. In other words, the literature has underrepresented important demographics from the global workforce, and the external validity of the findings presented thus far remain nebulous, compromising any efforts to use the findings reported in literature and to apply them ubiquitously across construction workers.

Another key gap in the body of knowledge is that researchers have found common stressors that impact construction workers in typical work environments using different assessment tools to measure stress levels. For example, through field experimentations, Goldenhar et al., 2003 [34] recognized ten work-related stressors that were directly related to safety outcomes: low job certainty, high work demands, low work control, poor safety climate, inadequate training, inability to apply skill(s), liability for other people's safety, poor safety compliance, long exposure hours, and years of experience. Other studies (e.g., [11,53,54]) found key subsets to these stressors that demonstrated the complexity caused by the lack of shared definitions of stress and stressors. Leung et al., 2010 [16] proposed that stressors influencing construction workers could be classified into five categories: (1) personal, (2) task, and (3) interpersonal, (4) organizational, and (5) physical. Factors such as work overload [16,17], poor physical environment [16,55], interpersonal conflict [56], poor organizational culture [18,57], and lack of appropriate safety equipment [16,57] can cause acute psychological and physical stresses that compromise the safe behavior and decision-making skills of workers [35,58,59]. All of the aforementioned works are seminal in their own respect; however, the challenge is that they only found synergies and not consistency in their assessment tools and results [12,34,60]. In sum, it remains unclear which of these stressors have high predictive capacity toward safety performance, enabling safety practitioners to prioritize resources and introduce targeted interventions.

To the best of our knowledge, 32 peer-reviewed studies were published in CEM literature between 1989 and 2020 that assessed and discussed both stress and stressors among construction workers. We have summarized the common stressors with definitions based on this comprehensive literature review. It should be noted that the example questions presented for each stressor reviewed in this section were typically asked in an interview setting or used in surveys with a Likert-styled scale, with the former allowing for qualitative explorations that provide epistemological truths and the latter allowing researchers to determine multivariate statistics that confirm or deny the significance of the different stressors. In some cases, to support the scope of this exploration, the adopted definition of a stressor subsumed some of the sub-stressors that were studied and discussed independently in the literature. Below, we discuss which sub-stressors were subsumed, with justification as to why the proposed model supported the objective of this study.

**Jobsite/Job Demand:** Job demand refers to the degree of cognitive and physical effort individuals perceive will be required of them to accomplish their work-related objectives [42]. This stressor appeared in 25 studies out of 32 (78.1%). Goldenhar et al., 2003 [34] collected data from over 400 workers that showed that job demand is directly correlated with poor safety performance. Job demand was noted to be a critical work-related stressor [61] with demands such as productivity pressures that can reduce vigilance towards hazards and risk management [62]. Here, we considered the following as indicators of job demand stressor: (1) work overload [60], (2) role overload [60], (3) emotional strain [15,35,39], (4) physical fatigue [63], (5) hours of exposure [34], (6) poor physical environment [63], (7) lack of goals [63], and (8) task demands [16]. Although some of these indicators have been occasionally used as independent stressors in past studies, for the purposes of this paper, the umbrella stressor of job demand captures work-execution challenges (e.g., work overload, role overload, lack of goal setting, and task demands), work-related exhaustion (e.g., emotional strain and physical fatigue), and safety concerns (e.g., hours of exposure and poor physical environment). Additionally, the use of these

different sub-stressors has been uneven in the literature, making it difficult to establish whether they are latent factors causing job-demand stress or stressors themselves. Finally, this categorization is also justified when one observes the types of questions asked to assess jobsite/job demand. For example:

> *"How often (do/did) you have to work very hard on the job?"* [34]

> *"How many hours per day are you exposed to each of the following hazardous or unpleasant conditions: noise, chemicals, asphalt, asbestos, and lead?"* [34]

> *"Temperature is too extreme"* [63]

**Job Control:** Job control refers to an individual's sense of autonomy in their job [16] and the authority to make decisions [64]. This translates to the latitude employees have in making decisions that may impact upon their tasks and their work environment and, in turn, on their motivation and self-efficacy [34,35]. This stressor appeared in 10 of the 32 studies reviewed (31.25%). As with job demand and role ambiguity, job control has been noted by researchers to negatively influence safety performance [34,35]. For the purposes of this paper, job control captures sub-stressors that relate to how much freedom and authority workers perceive or genuinely have within the work environment as it relates to their tasks, their co-workers, and their physical space, as there is a clear overlap among the constructs. This categorization is also justified when one observes the types of questions asked to assess job control. For example:

> *"I can control how fast I work"* [35]

> *"How much control (do/did) you have over how fast or slow you (work/worked)?"* [34]

> *"I was given insufficient authority to do my job properly"* [16]

**Job certainty:** Job certainty is an individual's sense of security and acceptance in their job [34]. Lack of certainty affects their confidence and ability to work, thereby causing work-related stress. The lack of certainty can prompt workers to take on risk and to make decisions that compromise both safety and job performance [16,34]. This stressor appeared in six of 32 studies reviewed (18.75%). One can justify this categorization by observing the kinds of questions asked to measure job certainty, such as the following:

> *"I am certain about the future of my job"* [35]

> *"How certain are you about your job future?"* [34]

**Interpersonal Conflicts at Work:** This stressor refers to adverse workplace interactions, ranging from insensitive conduct by co-workers to heated disputes. This stressor appeared in 10 of 32 studies reviewed (31.25%). We assumed that this stressor subsumes stressors such as workers/individual resilience [11], role conflict [60], conflict with co-workers [11], and poor communication [15]. This categorization is justified, given that the definition of these individual stressors is a subset of conflict in the workplace. Minimizing interpersonal conflicts in the work environment can yield a positive outlook among workers and build better coping mechanisms that prevent stress from other factors [11]. Beyond interactions among workers, ensuring that the roles of each individual are well-defined leads to fewer misunderstandings and better productivity. Role conflict can yield stress when there is a heightened sense of uncertainty and pressure to deliver the work [16]. These categorizations, under interpersonal conflicts, are supported by literature from analogous domains [65]. Resolving interpersonal conflicts not only leads to lower exposure to workplace aggression [66] and physical strain on workers [67], but also to significant labor and operational costs [68]. Some example questions used to assess this stressor include the following:

> *"How often do you get into arguments with your coworkers?"* [60]

> *"How often are your coworkers rude to you at work?"* [60]

**Role Ambiguity:** Role ambiguity refers to uncertainty concerning the nature of one's role in the workplace. If a person does not understand their duties and obligations, such

uncertainty can trigger high task-related negative pressure [16], which facilitates poor decisions that compound the stress experienced by the individual. Past explorations within CEM and analogous domains have found that role ambiguity among employees can negatively influence safety performance and learning outcomes [16,69]. This stressor appeared in four) of the 32 studies that were reviewed (12.5%). Example questions that were typically asked to assess this stressor included the following:

*"My goals and objectives are intangible and not clearly spelled out"* [16]

*"Explanations of what has to be done are often unclear"* [16]

**Skill Demand:** Skill demand is a stressor that results from a lack of the skill and experience required to carry out the work [60,70]. Skill demand also appeared in four of the 32 studies (12.5%). Unsurprisingly, skill demand has been found to negatively influence safety performance, as workers can experience frustration, added pressure, and/or perceive a lack of support in the workplace that prompts risk-taking behavior [34]. Here, we considered the following as indicators of skill demand: over-compensation [34], underutilization [70], and lack of safety knowledge [11]. This categorization is also justified when one observes the types of questions asked to assess jobsite/job demand; for example:

*"I would know what to do if an emergency occurred on my shift"* [11]

*"At work, how often (are/were) you are given a chance to do the things that would help you to improve or perfect your skills?"* [34]

*"How often on this job (do/did) you feel that you (have/had) to work harder than others in order to 'prove' yourself?"* [34]

Skill demand can yield negatively gendered impacts, with women perceiving and experiencing the need to demonstrate more skills than their male counterparts (i.e., over-compensation) to receive the respect and support within organizations [34]. On the other hand, the inability to showcase or to be personally challenged [70] or a lack of the requisite knowledge [11] can also induce stress among individuals who feel undervalued and incompetent, respectively. As these sub-stressors all relate to retention and to the demonstration of skills, considering skill demand as an overarching stressor is acceptable.

**Social Support:** Social support on jobsites refers to the relationships of an individual with peers, leaders, and the organization itself that can buffer the psychological impact of a high-stress job [71]. This stressor appeared five times in 32 studies (15.6%). Previous research discovered that social support can impact the overall level of job stress [34,35]. Increasing mental health-related concerns among workers have been hypothesized to be linked with the characteristics of the work environment and the workforce (e.g., toxic masculinity, physical demands, blame-based culture; [72]). Additionally, the advent and integration of robotics is challenging many of the traditional facets that make work experience positive: social interactions, movement, creativity, and autonomy [73]. Without support, workers can have negative work experience that could prompt risk-taking behaviors [74,75]. Some example questions used to assess this stressor include the following:

*"My co-workers contribute an extra effort to make my work life easier"* [35]

*"How often do your co-workers make an extra effort to make your work life easier for you?"* [34]

**Harassment and Discrimination**: Harassment and discrimination are unlawful actions that are perpetrated by individuals, organizations, communities, and governments. They include negatively targeting a particular person, gender, and/or community. Here, we defined both of these constructs only from the perspective of workplace and employment. In other words, discrimination is described as a collection of negative behaviors that establish social, psychological, and physical obstacles only for a particular individual or group of individuals [76]. According to the U.S. Equal Employment Opportunity Commission (EEOC), discrimination can be any unfair treatment, improper conduct, exclusion, or retaliation within a workplace environment [77]. Harassment, as defined by the EEOC, is

"unwelcome conduct that is based on race, color, religion, sex (including sexual orientation, gender identity, or pregnancy), national origin, older age (beginning at age 40), disability, or genetic information (including family medical history)." To treat someone less favorably or to engage in unwanted conduct can yield severe adverse outcomes in the physical and mental well-being of an individual [78–80]. For example, women and minorities are significantly more likely to experience discrimination and harassment at work through overt actions and microaggressions [34,81]. Unfair rewards and treatment are considered as indicators of harassment and discrimination [16], which may happen for a myriad of reasons. Some are due to the conditions around work (e.g., workers' limited role in organizations or an unforeseen decline due to projects' tight budgets) and others are due to malicious intent. Toxic working conditions can psychologically and physiologically impact workers; hence, it is unsurprising that the research showed that they have a negative impact on workers' overall performance, including safety [16,34,82]. This stressor was indicated seven times in 32 studies (21.8%). This categorization is also justified when one observes the types of questions asked to assess harassment and discrimination. For example:

*"I often feel that the organization treats us unfairly"* [16]

*"Have you ever felt that you were mistreated because you were a female/male by supervisors?"* [34]

**Supervisor Conflicts at Work:** This stressor refers to adverse interactions with workplace superiors, ranging from inconsistent treatment to disrespectful conduct [60]. Supervisors' roles in promoting safety and maintaining accountability are key to a healthy safety culture [24]. Conflicts in the workplace can create a hostile working environment that promotes aggression and results in stress. This stressor was indicated four times in 32 studies (12.5%). Example questions used to assess this stressor include the following:

*"How often do you get into arguments with your supervisors (subordinates)?"* [60]

*"How often are your supervisors (subordinates) rude to you at work?"* [60]

*"How often do your supervisors (subordinates) do mean things to you at work?"* [60]

**Job Satisfaction**: Job satisfaction is a function of balancing the rewards provided by the working environment and the person's preferences with respect to those rewards [83]. A lack of job satisfaction can be emotionally draining and distressing for individuals [84]. The construction industry isfacing an unprecedented shortage of skilled labor and is also struggling with high turnover and absenteeism. This stressor was indicated four times in 32 studies (12.5%). Some examples questions include the following:

*"All in all, how satisfied are you with your job?"* [85]

*"All in all, how satisfied are you with your company?"* [85]

While there has been substantial research aimed at understanding and assessing workplace stress among workers, it has been very disconnected in nature. The stressors included in the different assessment surveys are not consistent, which makes it challenging to compare and contrast the findings from different studies. This stifles any creative determination of actionable and targeted interventions to support safety leaders and practitioners. These inconsistencies in the assessment tools used by academics and practitioners stem from a lack of shared understanding as to which stressors have the most predictive capacity when predicting safety performance. For example, one study by Leung et al., 2012 [63] found a negative correlation between physical stress and accidents ($r = -0.107$), while another study by Leung et al., 2016 [35] found a negative correlation between physical stress and accidents ($r = -0.011$). However, whether these correlations are statistically significantly different or not has not been empirically established. Seminal work by different authors have not been in lock-step agreement with each other [34,52]. Different studies have also found different work-related psychological stressors to be relevant when correlated against safety performance. Siu et al., 2003 [15] identified two stressors, psychological distress ($r = 0.11$) and job satisfaction ($r = -0.14$); Leung et al., 2010 [16] found job stress ($r = 0.206$)

and emotional stress (0.254); Chen et al., 2017 [11] recognized three stressors: individual resilience ($r = -0.14$), interpersonal conflicts at work with supervisors ($r = 0.24$), and interpersonal conflicts at work with coworkers ($r = 0.26$). There is overlap between these stressors in how they are defined, but they are not defined so as to be interchangeable. This disparity is the reason there is a need for a formal meta-analysis to determine the true effect sizes of each work-related stressor identified in the literature, when predicting safety performance. This allows for the creation of a standardized assessment tool within CEM literature, while also supporting practitioners in prioritizing management strategies. This paper seeks to address this gap in the body of knowledge.

## 5. Performing a Meta-Analysis

Meta-analysis is the quantitative statistical approach through which findings from several studies can be synthesized to deliver holistic and aggregated findings [86]. A meta-analysis is a powerful analysis, as it allows us to leverage findings from different studies that on their own lack requisite sample sizes, scope limitations, and contextually apropos experimental and environmental changes to deliver findings with truly high external validity [87]. This approach allowed us to take seemingly incommensurable studies on the relationship between workplace stressors and safety performance to determine the salient stressors based on empirical evidence. This study follows the protocol described in the seminal work by Card, 2011 [25] to determine the relationship between work-related stressors and construction workers' injuries. This approach was also validated by Alruqi et al., 2018 [24] for the CEM context, where the authors used meta-analysis to identify key dimensions of safety climate and to correlate them with self-reported injury scores of construction workers. The analysis was performed in four unique stages (see Figure 1).

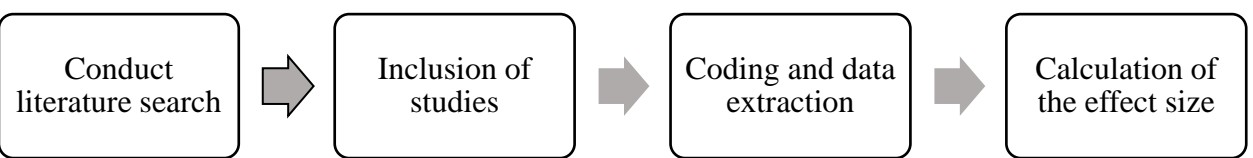

**Figure 1.** The described four stages for conducting meta-analysis.

## 6. Selecting Studies for Meta-Analysis

In the first stage, a comprehensive literature search was conducted to collect all studies investigating the relationship between construction-work-related stressors and construction workers' self-reported injuries. In this step, the literature review focused on studies that were only published within the CEM discipline. The main criteria for including a study in this meta-analysis was that the study investigated and reported sufficient information to conduct the meta-analysis, such as reporting the effect size for the relationship between work-related stressors and construction workers' injuries in the construction industry. Alternatively, information was provided in the published works to allow us to calculate the effect sizes. Finally, all selected papers were in English and published in a peer-reviewed journal. These criteria led to the inclusion of only seven out of the 98 studies found in the current published literature. As shown in Figure 2, this research used Excel software to arrange and filter searches from the websites. After extensive research on all articles related to the current study, 98 articles were included because they are about the relationship between psychological states and stress among construction employees. Subsequently, articles that did not meet the criteria were excluded. Finally, seven studies were selected, with respect to which the full criteria applied.

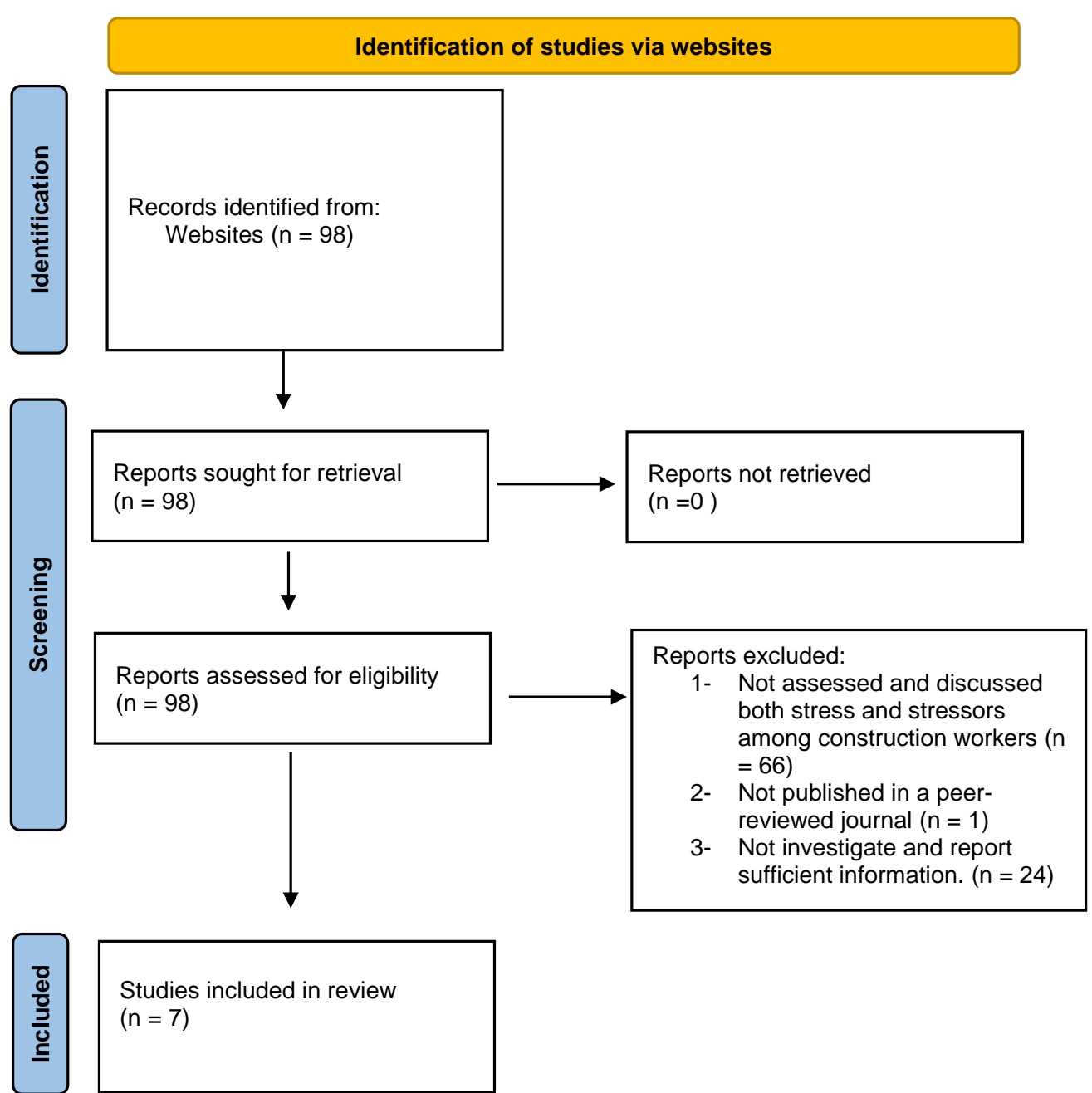

**Figure 2.** The PRISMA diagram for Grey Literature.

## 7. Coding

In the coding stage, each study was coded based on information such as category (e.g., workers or managers), study title, author, publication date, sample size, independent variables (i.e., construction stressor), and dependent variables (i.e., self-reported injury). The coding scheme for the studies included in this meta-analysis is shown in Table 1. The categorizations shown in Table 1, described in the previous section, allowed us to investigate independent stressors (i.e., to minimize overlap) against safety performance, while avoiding confounding effects.

**Table 1.** Studies included in the meta-analysis (*N* = 7).

| No. | Author (Data) | N | Independent Variables | Dependent Variable |
|---|---|---|---|---|
| 1 | Goldenhar et al., 2003 [34] | 408 | Job control. Job demands. Skill under-utilization Job certainty Overcompensating at work. Social support. Hours of exposure. Harassment and discrimination. | Self-reported injury |
| 2 | Chen et al., 2017 [11] | 837 | Individual resilience Interpersonal conflicts at work with co-workers | Self-reported injury |
| 3 | Chen et al., 2017 [60] | 837 | Work pressure. Role overload. Safety knowledge. Individual resilience. | Self-reported injury |
| 4 | Leung et al., 2010 [16] | 142 | Job stress. Emotional stress. | Self-reported injury |
| 5 | Leung et al., 2012 [63] | 395 | Emotional stress. Physical stress. Lack of goal setting Poor physical environment. Unfair reward and treatment | Self-reported injury |
| 6 | Leung et al., 2016 [35] | 166 | Job control Psychological stress Physical stress Job certainty Co-worker support | Self-reported injury |
| 7 | Siu et al., 2003 [15] | 374 | Psychological distress Communication | |

## 8. Overall Effect Size Computation

The final stage of the meta-analysis was to calculate the overall weighted effect size for each relationship. At this stage, we conducted several calculations to determine the overall effect size of the relationship between each stressor and injury. To do so, the individual effect sizes from each of the seven papers that met the established criteria for inclusion in meta-analysis were extracted. Then, the effect sizes were transferred to Fisher's exact test using Equation (1) to ensure that larger correlation values were not assigned a high weight just because they were large.

$$Z_r = 1/2 ln\left(\frac{1+r}{1-r}\right) \tag{1}$$

where *Zr* is Fisher's transformation of *r*, and *r* is the correlation value. The Fisher's *Z* values were then used to calculate the standard error for each study, using Equation (2):

$$SE_{Zr} = \left(\frac{1}{\sqrt{N-3}}\right) \tag{2}$$

The main assumption in the fixed effects model is that all effect sizes are similar in all included studies, whereas in the random effects model. the effect sizes differ from one study to another [88]. In this study, we used a random effects mode to estimate the overall effect size because it was assumed that each effect size in this study would be different. However, one required step before assigning weight to each study in the random effects model is testing the null hypothesis to ensure that there is no variation among the included effect sizes (i.e., a heterogeneity test), by using the information from the fixed effects model. (For more information on the fixed effects model calculation, please refer to Alruqi et al.,

2018 [24]). Equations (3) and (4) were used to calculate heterogeneity and the random variance of the included effect sizes.

$$Q = \sum \left( w_i \ ES_i^2 \right) - \frac{\left( \sum (w_i \ ES_i) \right)^2}{\sum w_i} \tag{3}$$

where $w_i$ is the weight of study $i$, $ESi$ is the effect size estimate for such a study in the fixed effect model, and $Q$ represents the heterogeneity statistic.

$$\tau^2 = \frac{Q - (k-1)}{\left( \sum w_i \right) - \frac{\left( \sum W_i^2 \right)}{\left( \sum W_i \right)}} \tag{4}$$

where $\tau^2$ represents random variance, $k - 1$ represents the degrees of freedom of $Q$, $Q$ is the heterogeneity statistic, $k$ is the number of included studies, and $w_i$ is the weight for each study. The random variance of the job certainty and social support stressors was assumed to be (0) because the result of the heterogeneity test for this relationship was less than the degree of freedom (i.e., $Q < k - 1$) [25]. Each effect size in the random effects model was assigned a weight, to allow a more accurate determination of the overall effect size using Equation (5):

$$w_i = \left( \frac{1}{SE_i^2} \right) \tag{5}$$

where $w_i$ is the weight for study $i$, and $SE_i$ is the standard error of the effect size estimate for study $i$.

Finally, the weighted mean effect size was calculated for each relationship, using the following equation:

$$\overline{ES} = \frac{\sum (w_i ES_i)}{\sum (w_i)} \tag{6}$$

where $w_i$ is the weight for study $i$, and $Es_i$ is the effect size calculated from individual studies.

However, to make the interpretation of the result easier, the overall effect size ($Zr$) was then converted to a correlation value, using Equation (7):

$$r = \left( \frac{e^{2z_r} - 1}{e^{2z_r} + 1} \right) \tag{7}$$

where ($r$) represents the correlation value and ($Zr$) represents Fisher's exact test value.

Using this validated approach (see Alruqi et al., 2018 [24]), we were able to leverage the findings from different studies in the CEM discipline that investigated the relationships between workplace stressors and injury rates to produce aggregated findings, as shown in Table 2.

**Table 2.** Correlation of the organization's stressors and construction workers' injuries.

| Stressor | $k$ | $(N)$ | $r$ | 95% CI *Zr* (LL) | 95% CI *Zr* (UL) | *p*-Value |
|---|---|---|---|---|---|---|
| Job Control | 2 | 574 | 0.16 | 0.02 | 0.31 | 0.014 |
| Job Demands | 13 | 5052 | 0.12 | 0.05 | 0.19 | 0.000 |
| Skill Demand | 3 | 1653 | 0.11 | 0.03 | 0.20 | 0.005 |
| Job Certainty | 2 | 574 | 0.11 | 0.03 | 0.19 | 0.004 |
| Social Support | 2 | 574 | 0.08 | −0.004 | 0.16 | 0.032 |
| Harassment and Discrimination | 2 | 803 | 0.11 | 0.02 | 0.21 | 0.007 |
| Interpersonal Conflicts at Work | 4 | 2877 | 0.17 | 0.10 | 0.24 | 0.000 |

Note: $k$ = number of studies (how many times that stressor was mentioned in the seven studies), $N$ = sample size, and $r$ = effect size of random effect model; 95% CI = confidence interval (lower–upper) around $r$.

## 9. Results

The inclusion criteria were met by seven of the 98 studies that were reviewed. These seven studies reported the required information to conduct the meta-analysis, such as the correlation between stress and self-reported injury. However, the meta-analysis's result identified a significant ($p < 0.05$) relationship between seven stressors and self-reported injury, as shown in Table 2.

Using the effect sizes (d = 0.2, 0.5, and 0.8) in benchmarks provided by Cohen, 2013 [89], one commonly accepted interpretation is that there are small (d = 0.2), medium (d = 0.5), and large (d = 0.8) effects, respectively. Therefore, all of the relationships between work-related stressors and injuries in this study had a low effect. The relationship between job control and injuries, for example, had a low effect ($r = 0.16$, 95% = 0.017 to 0.311). In addition, the relationship between job demand and injuries had a low effect ($r = 0.122$, 95% = 0.054 to 0.189). The effect of skill demand on injuries was also low ($r = 0.113$, 95% = 0.027 to 0.199), as was job certainty ($r = 0.11$, 95% = 0.028 to 0.192). In addition, social support, harassment and discrimination, and interpersonal conflicts at work had low effects ($r = 0.077$, 95% = −0.043 to 0.16; $r = 0.114$, 95% = 0.022 to 0.207; and $r = 0.17$, 95% = 0.096 to 0.244, respectively).

## 10. Discussion

The meta-analysis results indicated a significant relationship between seven stressors and self-reported injury: (1) interpersonal conflicts at work ($r = 0.17$); (2) job control ($r = 0.16$); (3) job demands ($r = 0.122$); (4) harassment and discrimination ($r = 0.114$); (5) skill demand ($r = 0.113$); (6) job certainty ($r = 0.110$); and (7) social support ($r = 0.077$). Role ambiguity, job satisfaction, and supervisor conflicts at work appeared in just one study. Consequently, it was not possible to perform a meta-analysis on those stressors.

Job demands were the most-mentioned stressor in the included studies, and the relationship between job demand and workers' self-reported injuries had a low effect ($r = 0.122$). That result of this meta-analysis was aligned with the results of the study by Chen et al., 2017 [60]. However, our result contrasted with the results in the study by Goldenhar et al., 2003 [34], which claimed that the relationship between job demands, and workers' injuries was not significant ($p = 0.25$, $p > 0.05$).

Job control was one of the fewest stressors mentioned; it was mentioned in two studies. The relationship between job control and injuries had a low effect ($r = 0.16$) and was significant ($p < 0.05$). Goldenhar et al., 2003 [34] found a negative correlation between construction workers' injuries and job control ($r = −0.1$), and it was significant ($p = 0.04$, $p < 0.05$), while Leung et al., 2016 [35] found a positive correlation between construction workers' accidents and job control (0.247), and it was of moderate effect. Those workers who have more decision-making power, such as the ability to choose their own work pace and whether to use protective equipment in some work that carries higher risk, have more control over their working conditions. Unfortunately, this could lead them to believe they are in control of the dangers and, as a result [90], ignore potential threats. This, in turn, increases the likelihood of accidents [35].

Job certainty was one of the stressors mentioned least; it was mentioned in two studies. The effect of job certainty was low ($r = 0.11$). The results align with the studies of both Goldenhar et al., 2003 [34] and Leung et al., 2016 [35] ($r = −0.08$, $r = −0.111$, respectively). Stressed-out construction workers are at risk of long-term health problems, due to the stresses they constantly deal with. In addition, their lack of job security could cause a long-term physiological change; that is, physical stress may be reactivated when the situation warrants it [35].

Skill demand was mentioned three times in the included studies and had three sub-stressors. The effect of skill demand on injuries was low ($r = 0.113$), and it was significant ($p = 0.005$, $p < 0.05$). The skill demand (safety knowledge) in the study by Chen et al., 2017 [60] was significant and had a negative correlation with injuries ($r = −0.18$). Employing a worker's skills more effectively, while simultaneously communicating that the crew, rather

than the individual worker, is accountable for the safety of co-workers, will help workers reduce their psychological issues, thus reducing the occurrence of near misses [34].

Social support was mentioned two times in the included studies. It had a low effect ($r = 0.08$), and it was significant ($p = 0.03$, $p < 0.05$) in the current study. In contrast, the social support stressor was not significant in the studies of both Goldenhar et al., 2003 [34] and Leung et al., 2016 [35]. Furthermore, workers should be placed in teams within which they will regularly interact with other workers, such as structural steel erectors and structural steel welders, to help each other and support each other in adhering to safe practices [35]. Having co-workers' support allows construction workers to do their work more efficiently and reduces the risk of stress-related illnesses [91,92].

The harassment and discrimination stressor had a low effect ($r = 0.11$), and it was significant ($p = 0.007$, $p < 0.05$) in this study. In contrast, the harassment and discrimination stressor was not significant in the studies of both Goldenhar et al., 2003 [34] and Leung et al., 2012 [63]. It was mentioned twice in the included studies. Women face discrimination and sexual harassment in the workplace more than men, especially in traditionally male-dominated fields, such as construction. The abuse of power has harmed well-being in physical, psychological, and job-related ways. In addition, harassment and discrimination lead to more near misses and injuries, which may, in turn, cause stress and hypertension. Thus, it is terrible for the workers and harms the construction industry [34].

Interpersonal conflict at work had a low effect, and it was significant ($r = 0.17$, $p < 0.05$). In comparison, it was not significant in the study of Siu et al., 2003 [15] (communication). It was mentioned four times in the included studies, and it has three sub-stressors. Workplace training can include improving employees' coping abilities and individual resilience to help them deal with interpersonal conflict, which could help prevent and minimize interpersonal conflicts at work [60].

The current emphasis on workplace safety and health has gone from being regarded as acceptable to being regarded as necessary. Researchers have begun to explore the connection between stressors and worker performance/mental health issues [11,12]. Stressors such as work overload [16] and interpersonal conflict [56] can place employees at risk in critical situations, such as falling, being struck by objects, getting shocked, or being hurt if caught in machinery [35,58,59].

## 11. Limitations

The small number of studies that met our inclusion criteria and the small sample sizes were the main limitation of this meta-analysis study. Of the 98 studies on stress in the construction industry, 16 focused on construction workers' injuries. However, nine studies were excluded because of the lack of sufficient statistical information to calculate effect sizes. In the future, researchers need to consider publishing or making available all results related to the relationship between work-related stressors and construction workers' injuries, to have a larger sample. The limitation of small sample size also prevented the analysis of the relationship between many stressors and construction workers' injuries, such as supervisor conflicts at work, role ambiguity, and job satisfaction.

## 12. Conclusions

This paper presented an analysis of the relationship between work-related stressors and construction workers' injuries, with the aim of finding the essential work-related stressors that affect construction workers' safety. While other researchers have investigated the relationship between work-related stressors and construction workers' injuries, this paper presents the first formal meta-analysis of this significant research issue. Our study identified jobsite demand, job control, job security, skill demand, social support, harassment and discrimination, and interpersonal conflicts at work as major stressors that are significantly correlated with worksite injuries ($p < 0.05$). The effect sizes of the relationships between the seven stressors and injuries were low. The small number of studies that met our inclusion criteria, the small sample sizes, and the lack of sufficient statistical information to calculate

effect sizes were the main limitations of this meta-analysis study. However, the results presented in this study are an essential first step in determining the work-related stressors that impact construction workers and encouraging researchers to collect consistent and reliable data.

**Author Contributions:** Conceptualization, B.M.A., W.A. and S.B.; methodology, B.M.A., W.A., S.B., O.A. and H.L.; software, B.M.A., W.A. and S.B.; validation, B.M.A., W.A., S.B., O.A. and H.L.; formal analysis, B.M.A. and W.A.; investigation, B.M.A., W.A., S.B., O.A. and H.L.; data curation, B.M.A. and W.A.; writing—original draft preparation, B.M.A.; writing—review and editing, B.M.A., W.A., S.B., O.A. and H.L.; supervision, W.A., S.B., O.A. and H.L.; project administration, S.B. and O.A. All authors have read and agreed to the published version of the manuscript.

**Funding:** This research received no external funding.

**Data Availability Statement:** Some or all of the data, models, or codes that support the findings of this study are available from the corresponding author upon reasonable request.

**Conflicts of Interest:** The authors declare no conflict of interest.

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
