# Peer review of "The Relationship between Work-Related Stressors and Construction Workers’ Self-Reported Injuries: A Meta-Analytic Review"

_2673-4109, doi:10.3390/civileng3040062_

Round 1
Reviewer 1 Report
Please add a description of the inclusion and exclusion criteria used to identify relevant studies
Please add a PRISMA diagram to illustrate the selection process for this review
Please add a description of study characteristics
Please comment on the implications of a low effect size for all identified stressors and workplace injury despite producing a significant p value
Minor grammatical errors are indicated on the attached file

Author Response
Dear reviewer,
Thank you for your honest and helpful review. In the file are the responses to all the points you asked about.
All grammatical errors have also been corrected.
Thank you very much.

Reviewer 2 Report
This is an important topic and a sophisticated review and analysis were provided by the authors. Overall, this is a quality research study that yields empirical findings across seven other source studies that were standardized and summarized in a meta-analysis procedure.
I found the paper clear and easy to understand. The references cited cover the main types of stressors for construction workers from the workplace context. The results obtained make sense with past studies and the interpretations of the findings follow the data.
Minor edits to consider:
The Results section should not present the r values and p values of findings in Table 1 - as this is repetitive.
The tables and text do not need a 0 before the period for correlation values and probability significance values. Please be consistent in using a .xx or .xxx level of detail in the correlations reported.
Some of the references are not fully done in the style of the journal. Please edit all references as needed. Most missing the online address needed to access the paper - either DOI number or the specific full website address link.
Although the paper does a thorough job of describing and citing the major conceptual domains of risk factors and working conditions that are associated with worker on-the-job injuries, the personal level psychological and mental health / behavioral health influences could be discussed more. More specifically, the role of employee assistance programs (EAP) in supporting at-risk construction workers with counseling and risk assessment is well-established (yet EAP is not mentioned in the study). See an overview of EAP in the USA here:
https://journals.sagepub.com/doi/pdf/10.1177/08901171221112488d
See some recent review resources below from Cal Beyer and others to consider that focus on construction industry issues:
http://hdl.handle.net/10713/16808
http://hdl.handle.net/10713/12703 http://hdl.handle.net/10713/17304http://hdl.handle.net/10713/15522
http://hdl.handle.net/10713/16758
http://hdl.handle.net/10713/16759
http://hdl.handle.net/10713/16762
http://hdl.handle.net/10713/16786
Author Response

(The authors gave the same response as above.)

Reviewer 3 Report
I read with great interest the Manuscript titled “The Relationship Between Work-Related Stressors and Construction Workers' Self-Reported Injuries: A Meta-Analytic Review”, which falls within the aim of Advances in Civil Engineering.
In my honest opinion, the topic is interesting enough to attract the readers’ attention. Moreover, the methodology is accurate, and conclusions are supported by the reported data. Nevertheless, the authors should improve further the Manuscript.
TITLE
The title of the article is accurate.
ABSTRACT
Some clarifications are however needed. Population, Intervention, Comparison, Outcomes and Study (PICOS) design must be used as a framework to formulate eligibility criteria in Meta‐Analysis / systematic review. The Abstract should be rewritten to include the PICOS (population, intervention, comparison, outcome, and study design) of the review. The same should be added at the end of the introduction before Materials & Methods.
INTRODUCTION
The introduction is coherent. It also creates a niche for the current investigation by showing a hole in the literature and discussing how it intends to fill it. However, sentence about PICOS (population, intervention, comparison, outcome, and study design) should be added at the end of the introduction before Materials & Methods.
METHOD
Some clarifications are however needed.
Was this review registered in PROSPERO, the Registry of Systematic Reviews/Meta-Analyses in Research Registry, or in INPLASY, which are specific to systematic reviews? Unfortunately, I could not find any information about this point.
Why “However“ is repeated two times in line 478?
RESULTS
The technique of data analyses seems appropriate but the results section is missing some information. Please report information about publication bias (eg., symmetry of the funnel plot) and in case of asymmetry report the results of Duval and Tweedie's Trim and Fill method.
TO SUM UP I think the author(s) need to make the recommended corrections.
Author Response

(The authors gave the same response as above.)
